

**Measurement report: Long emission-wavelength chromophores dominate the light absorption**
**of brown carbon in Aerosols over Bangkok: impact from biomass burning**
Jiao Tang[1,2,3], Jiaqi Wang[1,2,3,7], Guangcai Zhong[1,2,3], Hongxing Jiang[1,2,3,7], Yangzhi Mo[1,2,3], Bolong
Zhang[1,2,3,7], Xiaofei Geng[1,2,3,7], Yingjun Chen[4], Jianhui Tang[5], Congguo Tian[5], Surat Bualert[6], Jun
Li[1,2,3], Gan Zhang[1,2,3]
[1]State Key Laboratory of Organic Geochemistry and Guangdong Key Laboratory of Environmental
Protection and Resources Utilization, Guangzhou Institute of Geochemistry, Chinese Academy of
Sciences, Guangzhou 510640, China
[2]CAS Center for Excellence in Deep Earth Science, Guangzhou, 510640, China
[3]Joint Laboratory of the Guangdong-Hong Kong-Macao Greater Bay Area for the Environment,
Guangzhou Institute of Geochemistry, Chinese Academy of Sciences, Guangzhou 510640, China
[4]Department of Environmental Science and Engineering, Fudan University, Shanghai 200092, P.R.
China
[5]Key Laboratory of Coastal Environmental Processes and Ecological Remediation, Yantai Institute of
Coastal Zone Research, Chinese Academy of Sciences, Yantai 264003, China
[6]Faculty of Environment, Kasetsart University, Bangkok, 10900, Thailand
[7]University of Chinese Academy of Sciences, Beijing 100049, China
**Correspondence:** Guangcai Zhong (gczhong@gig.ac.cn)



**Abstract:** Chromophores represent an important portion of light-absorbing species, i.e. brown carbon.
Yet knowledge on what and how chromophores contribute to aerosol light absorption is still sparse. To
address this problem, we examined soluble independent chromophores in a set of year-round aerosol
samples from Bangkok. The water-soluble chromophores identified via excitation-emission matrix
(EEM) spectroscopy and follow-up parallel factor analysis could be mainly assigned as humic-like
substances and protein-like substances, which differed in their EEM pattern from that of the methanol-
soluble fraction. The emission wavelength of chromophores in environmental samples tended to
increase compared with that of the primary combustion emission, which could be attributed to
secondary formation or the aging process. Fluorescent indices inferred that these light-absorbing
chromophores were not significantly humified and comprised a mixture of organic matter of terrestrial
and microbial origin, while these inferences exhibited a refutation with primary biomass burning and
coal combustion results. A multiple linear regression analysis revealed that larger chromophores that
were oxygen-rich and highly aromatic with high molecular weights, were the key contributors of light
absorption, preferably at longer emission wavelength ($\lambda_{max} > 500$ nm). Positive matrix factorization
analysis further suggested that up to 60% of these responsible chromophores originated from biomass
burning emissions.





## 1. Introduction

Atmospheric aerosols play a substantial role in climate change through radiative forcing (Alexander et al., 2008). Carbonaceous aerosols mainly include organic carbon (OC) and elemental carbon (EC). Brown carbon (BrC) is a specific type of OC that absorbs radiation efficiently in the near-ultraviolet and visible (UV-vis) range (Laskin et al., 2015;Kirchstetter et al., 2004) and may contribute 15% or more of total light absorption over the UV-vis spectrum (Kirchstetter and Thatcher, 2012;Liu et al., 2013). This fraction can significantly affect atmospheric chemistry, air quality, and climate change (Marrero-Ortiz et al., 2018;Laskin et al., 2015). Forest fires, residential heating by wood and coal, biogenic release, and secondary formation contribute to BrC in the atmosphere (Laskin et al., 2015). Many studies have indicated that the optical properties of BrC may significantly evolve as a result of atmospheric processes such as oxidation (Fan et al., 2020), solar irradiation (Wong et al., 2017), and relative humidity (Kasthuriarachchi et al., 2020). These factors cause variability in the chemical compositions and levels of BrC across source regions and receptors, resulting in a high degree of uncertainty regarding the effects of BrC (Dasari et al., 2019;Xie et al., 2019).

Light absorption of BrC is associated with its molecular composition and chemical structure (Song et al., 2019;Lin et al., 2018;Mo et al., 2018;Jiang et al., 2020). Detailed structural characterization of BrC compounds is essential to understand their sources and chemical processes in the atmosphere. High-resolution mass spectrometry (HRMS) is a powerful tool for molecular-level chemical analysis of organic aerosols (Laskin et al., 2018). Combinations of offline high-performance liquid chromatography (HPLC), a photodiode array detector, and HRMS allow the chemical characterization of aerosols specific to BrC (Lin et al., 2018;Lin et al., 2016;Lin et al., 2015;Lin et al., 2017). With these combination approaches, nitroaromatics, aromatic acids, phenols, polycyclic aromatic hydrocarbons and their derivatives are basically identified as BrC chromophores (Wang et al., 2020b;Yan et al., 2020). However, it should be noted that it is difficult to ionize some organic compounds for detection using HRMS, and even for those that can be detected, HRMS can only provide possible molecular structures based on empirical deduction (Song et al., 2018;Lin et al., 2015). The isomeric complexity of natural organic matter may have exceeded achievable one-dimensional chromatographic resolution (Hawkes et al., 2018), and therefore, the majority of components in the BrC mixture remain undetermined.

Excitation-emission matrix (EEM) fluorescence spectroscopy detects bulk chromophores in a solution (Chen et al., 2016). Chromophores can be revealed by EEM with information on their chemical structures associated with molecular weight, aromatic rings, conjugated systems, etc (Wu et al., 2003). For example, a redshift in emission spectral maxima can be caused by an increase in the





number of aromatic rings condensed in a straight chain, conjugated double bonds, or formational
changes that permit vibrational energy losses of the promoted electrons (Wu et al., 2003). A significant
Stokes shift with emission wavelength can be observed in aged secondary organic aerosols (SOA)
using EEM spectroscopy (Lee et al., 2013). Parallel factor (PARAFAC) analysis has been widely used
to decompose the EEM spectral signature into independent underlying components (Han et al.,
2020;Yue et al., 2019;Wu et al., 2019;Chen et al., 2019b), adding valuable information to absorbance-
based measurements (Yan and Kim, 2017). This technique helps to categorize groups of similar
fluorophores or chromophores or similar optical properties, thereby allowing a better understanding of
the chemical properties of BrC. There is evidence that BrC absorption is closely correlated with
chromophores (Huo et al., 2018). However, the intrinsic relationship between chromophores and BrC
absorption has not been explored.
Southeast Asia is subject to intensive regional biomass burning, the emissions from which may
contribute to atmospheric brown clouds (Ramanathan et al., 2007;Laskin et al., 2015). The contribution
of biomass burning to aerosol optical depth was evaluated to be more than 56% over this region (Huang
et al., 2013). Despite many studies focused on the characterization of atmospheric black carbon (BC)
(See et al., 2006;Fujii et al., 2014;Permadi et al., 2018), studies on BrC in the region are still limited.
A recent study in Singapore indicated that water-soluble OC (WSOC) exhibited strong wavelength
dependence and even higher values of BrC absorption than those from Korea, India, China, and Nepal
(Adam et al., 2020), indicating abundant water-soluble BrC in the air over Southeast Asia.
This study was performed to explore the relationships between EEM chromophores and BrC light
absorption in soluble aerosol organic matter. A set of year-round aerosol samples from Bangkok,
Thailand, was analyzed. Water-soluble and methanol-soluble BrC in the aerosol samples were
characterized by EEM followed by statistical analyses to retrieve information on the contributions of
fluorescent chromophores to BrC light absorption, as well as their emission sources. This study
provides a comprehensive dataset on seasonal variability in the light absorption properties, sources,
and chemical components of BrC, which may be useful for improving further modeling and field
observation.
**2. Experiment**
**2.1. Sample Collection and Extraction.**
Eighty-five total suspended particulate (TSP) samples were collected on the roof (57 m above
ground level) of the Faculty of Environment, Kasetsart University (100°57′ E and 13°85′ N) in
Bangkok, Thailand (Fig. S1). Detailed information about the sampling site is presented elsewhere
(Wang et al., 2020a). Sampling was performed from January 18, 2016 to January 28, 2017, and the





sampling period was divided into four seasons: the pre-hot season (January 18–February 28, 2016),
hot season (March 2–May 30, 2016), monsoon (June 2–October 30, 2016), and cool season (November
1, 2016–January 28, 2017). Table S1 lists the average meteorological data in the four seasons.
Generally, during the sampling period, the hot season was characterized by high temperatures and wind
speeds, and the monsoon season by high humidity. TSP samples were collected over 24 h using a high-
volume ($0.3 \, m^3 \, min^{-1}$) sampler with quartz-fiber filters (QFFs, prebaked for 6 h at 450 °C). All samples
and field blanks were stored under dark conditions at –20 °C until analysis.

WSOC was prepared by ultrasonication extraction of filter punches with ultra-pure deionized

water ($\Omega > 18.2$). The methanol-soluble OC (MSOC) fraction was then obtained by extracting the
freeze-dried residue on GFFs with HPLC-grade methanol, which is used for water-insoluble fractions
(Chen and Bond, 2010). The extract solutions were passed through 0.22-μm PTFE filters and subjected
to follow-up UV-vis absorption and fluorescence spectral analysis. The mass concentrations of WSOC
and MSOC were measured, and the method are shown in the Supplement.
**2.2. Absorption Spectra and Fluorescence Spectra.**

The extract solutions were placed in quartz cells with a path-length of 1 cm and subjected to

analysis using an fluorometer (Aqualog; Horiba Scientific, USA). Absorption spectra and EEM spectra
were obtained simultaneously using this instrument. The contribution of solvents was subtracted from
the extract spectra. UV-vis absorption spectra were scanned in the range of 239 to 800 nm with a step
size of 3 nm. The Fluorescence spectra were recorded with emission wavelength (Em) ranging from
247.01 to 825.03 nm and excitation wavelength (Ex) ranging from 239 to 800 nm. The wavelength
increments of the scans for Em and Ex were 4.66 and 3 nm, respectively. The calculation of optical
parameters and the relative contributions of BrC to total aerosol light absorption are presented in the
Supplement.
**2.3. Factor analysis**

In this study, we built a PARAFAC model, based on 85 TSP sample fluorescence (samples × Ex

× Em: 85 × 188 × 125, 85-model). Original EEM spectra were corrected and decomposed via
PARAFAC analysis with reference to earlier methods using drEEM toolbox version 2.0 with MATLAB
software (http://models.life.ku.dk/drEEM, last access: June 2014) (Murphy et al., 2013;Andersson and
Bro, 2000). The absorbance, all below 1 at 239 nm, was deemed suitable for correcting the EEM
spectra for inner filter effects (IFEs) (Luciani et al., 2009;Gu and Kenny, 2009;Fu et al., 2015), and the
sample EEM spectra, and blanks were normalized relative to the Raman peak area of ultrapure
deionized water collected on the same day to correct fluorescence in Raman Units (RU) (Murphy et




al., 2013;Murphy et al., 2010). Spectra with Em > 580 nm and Ex < 250 nm were removed to eliminate
noisy data. The non-negativity constraint is necessary to obtain reasonable spectra, and signals of first-
order Rayleigh, Raman, and second-order Rayleigh scattering in the EEM spectra were removed using
the interpolation method (Bahram et al., 2006). The two- to nine-component PARAFAC model was
explored, within the context of spectral loading, core consistency, and residual analysis (Figs. S2–S5).
Finally, seven and six components were identified in the WSOC and MSOC fractions, which explained
99.89% and 99.76% of the variance, respectively. Both the seven- and six-component PARAFAC
solutions passed the split-half analysis with the split style of "$S_4C_6T_3$", and residuals were examined
to ensure that there was no systematic variation. The parameters obtained from the PARAFAC model
were used to calculate the approximate abundance of each component, expressed as $F_{max}$ (in RU),
corresponding to the maximum fluorescence intensity for a particular sample.
Fluorescence indices based on intensity ratios that provide insight into the origins of dissolved
BrC, such as the humification index (HIX) (the ratio of average emission intensity in the 435−480-nm
range to that in the 300−345-nm range following excitation at 254 nm, which was used to reflect the
degree of humification) (Zsolnay et al., 1999), the biological index (BIX) (the ratio of emission
intensities at 380 and 430 nm following excitation at 310 nm, reflecting autochthonous biological
activity in water samples) (Huguet et al., 2009), and fluorescence index (FI) (the ratio of emission
intensities at 470 and 520 nm following excitation at 370 nm, reflecting the possibility of microbial
origin and for examining differences in precursor organic materials) (Lee et al., 2013;Murphy et al.,

2018).

**2.4. Statistical analysis**

A hierarchical cluster method was used to classify aerosol samples based on the relative
contributions of PARAFAC components to the respective samples. The Squared Euclidean distance
method was used to evaluate the distances between samples, and the Between-group linkage method
was chosen for hierarchical cluster analysis. The multiple linear regression (MLR) model was applied
to elucidate the relationship between chromophores and light absorption of BrC using a stepwise
screening process. Analyses were performed using SPSS software (SPSS Inc., Chicago, IL, USA).

**3.   Results and Discussion**

**3.1. EEM of dissolved organic substances.**

Fluorescence spectra coupled with PARAFAC results can provide more information about the
chemical structures of chromophores. Figure 1 and Table S2 show the seven-component (P1–7)
PARAFAC solutions of WSOC in the samples of aerosol over Bangkok, the peaks of which fell mainly





into the humic-like and protein-like chromophore regions in the plots. Components P2, P3, P4, and P6
were identified as humic-like substances (HULIS) (Chen et al., 2017a;Stedmon and Markager,
2005;Wu et al., 2019;Chen et al., 2003). A second peak was observed at a high excitation wavelength
for these components, indicating the existence of a large number of condensed aromatic moieties,
conjugated bonds, and nonlinear ring systems (Matos et al., 2015). Among them, P2, P3, and P4 had a
longer emission wavelength (> 400 nm) than P6, likely due to the low probability of fluorescence
emission from quinonoid $n\text{-}\pi^*$ transitions (Cory and McKnight, 2005). P3 produced similar spectra to
those of aqueous reaction products of hydroxyacetone with glycine (Gao and Zhang, 2018), and
dissolved organic matter (DOM) in the surface water of Xiangxi Bay and Three Gorges Reservoir
(Wang et al., 2019). P6 had a peak similar to those in the fluorescence spectra of N-containing SOA
species formed by α-pinene under ozonolysis and photooxidation with $NH_3$ in a flow reactor (Babar et
al., 2017) as well as pyridoxine (Pohlker et al., 2012), indicating a possible biological source. P5 was
similar to a previously identified fluorophore in $PM_{2.5}$ from Xi'an (Chen et al., 2019b). P1 and P7
could be assigned as protein-like organic matter (PLOM) due to their short emission wavelengths (Wu
et al., 2003). Specifically, P7 resembled a tyrosine-like fluorophore (Zhou et al., 2019;Chen et al., 2003)
and may be related to non-N-containing species (Chen et al., 2016).
The MSOC fraction extracted from the filter residue after water extraction produced fluorescence
signals with fluorescence patterns different from those of the WSOC fraction, indicating a different
chemical composition from that of WSOC. Thus, WSOC with the addition of MSOC may provide a
more comprehensive description of the optical and chemical characteristics of BrC compared to
WSOC alone. Six components (C1–C6) were resolved for the MSOC. Among them, C1 and C2 were
associated with shorter excitation wavelengths (< 250 nm) but longer emission wavelengths (> 380
nm), indicating the presence of fulvic-like substances (Chen et al., 2003;Mounier et al., 1999). C6
produced a pattern similar to that of tyrosine-like fluorescence (Stedmon and Markager, 2005).
Although C4 had a similar EEM spectrum as P4 of WSOC, the two components were chemically
different in polarity, suggesting different behaviors in the environment (Ishii and Boyer, 2012). Note
that there were no special chemical structures for the different types of chromophores, and therefore,
the origins and chemical structures of HULIS and PLOM studied here are not necessarily like those
with the same names in other types of organic matter.

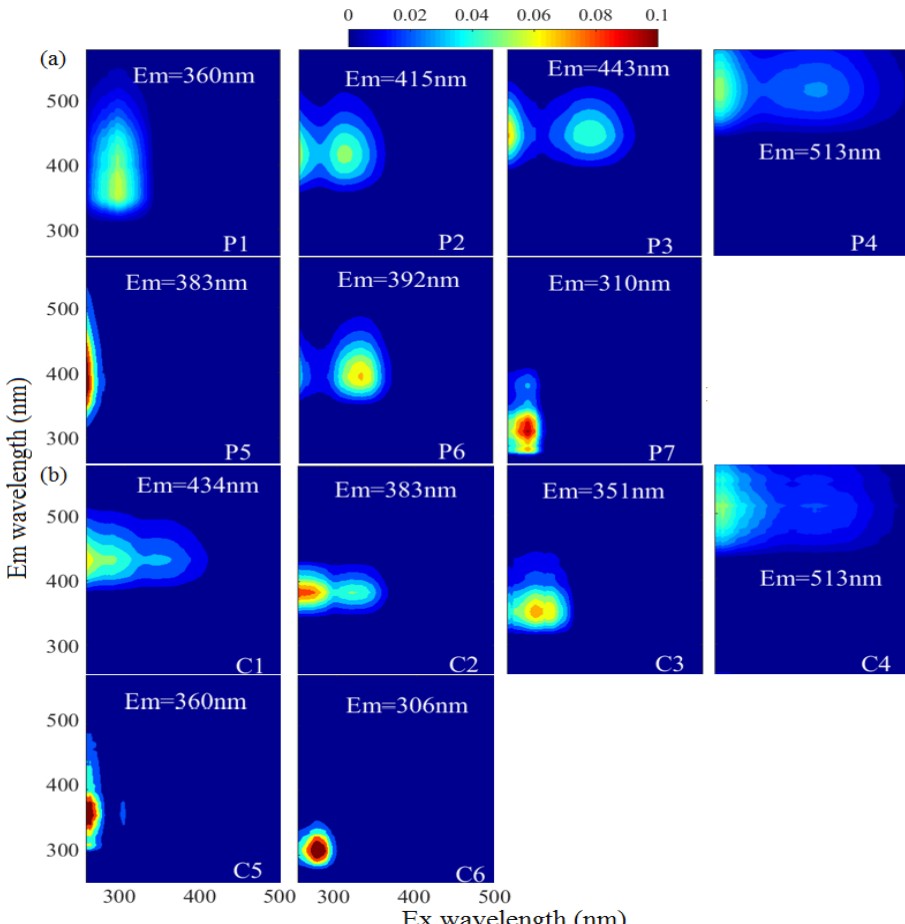

**Figure 1.** The fluorescent components identified by the PARAFAC (parallel factor) analysis for EEM of water-soluble organic carbon (P1−P7, WSOC, a) and methanol-soluble organic carbon (C1−C6, MSOC, b) in the aerosol samples over Bangkok in Thailand (n=85).

To further explore the potential sources of the EEM-PARAFAC components, we added 60 source samples to the matrices. The source sample EEM data were described in our previous study (Tang et al., 2020b), including those of 33 biomass-burning samples (IDs: 1–33), 17 coal-combustion samples (IDs: 34–50) samples, eight tunnel samples (IDs: 51–58) and two vehicle-exhaust samples from trucks (IDs: 59–60) , which are important sources of BrC in the atmosphere. This, in combination with our Bangkok field samples, yielded a new matrix (145 × 188 × 125, 145-model) for modeling. PARAFAC analysis successfully decomposed the dataset, and the output was the same as for the 85-model. The component solutions are presented in Fig. S6. To validate the stability of the model after loading by the new matrix, the Tucker congruence coefficient (TCC) was calculated to determine the similarity



of two fluorescence spectra between the two models (refer to Text S3 of Supplement). Note that a
higher TCC value would indicate a higher degree of similarity of the spectra. As shown in Table S2
and Fig. S7, high TCC values were found as expected between the 85-model components and the 145-
model components, indicating that the two models identified similar chromophores. Although one
exceptional component was detected each for the WSOC and MSOC fractions by the new 145-model,
these fluorescent components were only highly characterized by source samples, as reported in our
previous study (Tang et al., 2020b).

Using the distribution proportions of the EEM-PARAFAC fitted components (145-model), we

conducted hierarchical cluster analysis of the mixed ambient and source samples. The results are
shown in Figs. S9 and S10. For the WSOC fraction, all aerosol samples from Bangkok and tunnel
samples were assigned to cluster A, whereas biomass-burning and coal-combustion aerosols were
assigned to clusters C and D, respectively. This implied that the chromophore types could be somewhat
related to the emission precursors of the aerosol components. However, the distribution of
chromophores varied clearly between the ambient aerosols and source samples. The ambient aerosol
samples contained higher levels of chromophores with longer emission wavelengths that were related
to humic-like or fulvic-like chromophores (components 145M-P1 (P1 component in 145-model),
145M-P5, and 145M-P6), whereas the primary biomass-burning and coal-combustion samples
contained high-intensity chromophores with shorter emission wavelengths that were related to protein-
like fluorescence (145M-P2 and 145M-P4). These phenomena was similarly reported previously, i.e.,
protein-like substances produce compounds with similar fluorescence properties as humic substances
under irradiation conditions (Bianco et al., 2014). Similar differences between field samples and source
samples were found for the MSOC fraction. Therefore, our results confirmed that chemical reactions
or "aging" in the atmosphere greatly modifies the chromophore patterns of emission sources by both
bleaching the source chromophores or producing new chromophores and, at least in this case, shifts
the chromophore emission wavelength toward longer wavelengths, i.e., from protein-like to fulvic-like
(Bianco et al., 2014;Bianco et al., 2016;Lee et al., 2013).
**3.2. Fluorescence-derived indices**

The ratios of fluorescence intensity from specific spectral regions of an EEM were used as

indicators for the relative contributions of organic matter derived from terrestrial or microbial sources
in natural waters (Shimabuku et al., 2017;Birdwell and Engel, 2010;Mcknight et al., 2001). HIX was
initially introduced to estimate the degree of maturation of DOM in soil (Zsolnay et al., 1999),
representing the degree of humification of organic matter, for which higher HIX values also indicate
higher degree of polycondensation (low H/C ratio) and aromaticity (Qin et al., 2018). Generally, high





HIX values (> 10) correspond to strongly humified or aromatic organics, principally of terrestrial
origin, whereas low values (< 4) are indicative of autochthonous or microbial origin. As shown in
Table 1 and Fig. 2, the HIX values were 3.4±0.99 and 2.0±0.59 for WSOC and MSOC, respectively,
in aerosol samples from Bangkok. All HIX values were less than 10, which could be viewed as a
nominal cutoff below which DOM is not significantly humified (Birdwell and Valsaraj, 2010;Zsolnay
et al., 1999;Huguet et al., 2009). Figure 2 shows the HIX values in primary biomass-burning and coal-
combustion samples, which were much lower than those in the ambient samples, indicating that the
lower values of HIX in the atmosphere likely correspond to freshly introduced material. Lee et al.
(2013) reported that fresh SOA had low HIX values, but these values increased significantly upon
aging with ammonia. The much higher HIX values in the WSOC compared to the MSOC suggest that
WSOC may have a higher degree of aromaticity or a more condensed chemical structure. Our previous
study revealed that MSOC has a higher molecular weight but lower aromaticity index than the
corresponding WSOC in combustion experiment aerosol samples, indicating a more aliphatic structure
in the MSOC (Tang et al., 2020b). The HIX values of WSOC were highest in the hot season (3.9±1.1),
followed by the pre-hot season (3.3±1.1), cool season (2.9±0.36), and monsoon (2.5±0.22), whereas
those of the MSOC tended to be higher in the hot and cool seasons than in the monsoon and pre-hot
seasons. The HIX values in the WSOC fraction were comparable to those of water-soluble organic
aerosols in the high Arctic atmosphere (mean: 2.9) (Fu et al., 2015) and higher than those of water-
soluble aerosols (1.2±0.1 in winter and 2.0±0.3 in summer) over northwest China (Qin et al., 2018),
likely indicating a higher degree of chromophore humification.
**Table 1** Seasonal averages of the concentration of organic carbon (OC), elemental carbon (EC), water-soluble organic
carbon (WSOC), and methanol-soluble organic carbon (MSOC), BrC absorption, fluorescence indices and
levoglucosan level for aerosol samples collected from Bangkok in Thailand. Pre-hot season is from January 18 to
February 29, 2016; hot season is from March 2 to May 31, 2016; monsoon is from June 2 to October 30, 2016; cool
season is from November 1, 2016 to January 28, 2017.

| | Annual (n=85) Ave ± sd | Pre-Hot season (n=7) Ave ± sd | Hot season (n=41) Ave ± sd | Monsoon (n=7) Ave ± sd | Cool season (n=30) Ave ± sd |
|---|---|---|---|---|---|
| [a]OC ($\mu$g C m$^{-3}$) | 12±7.3 | 19±9.3 | 9.6±6.7 | 6.5±0.97 | 16±5.6 |
| [a]EC ($\mu$g C m$^{-3}$) | 1.4±0.48 | 2.0±0.45 | 1.2±0.47 | 1.2±0.15 | 1.5±0.40 |
| [a]OC/EC | 8.9±5.2 | 9.6±3.4 | 8.4±6.8 | 5.4±0.51 | 10±2.5 |
| | | | WSOC | | |
| $\mu$g C m$^{-3}$ | 6.2±4.2 | 9.9±5.7 | 5.3±4.1 | 2.6±0.31 | 7.4±3.4 |
| AAE (330–400 nm) | 5.1±0.68 | 5.0±0.52 | 5.4±0.56 | 6.2±0.11 | 4.5±0.34 |


| | | | | | |
|---|---|---|---|---|---|
| Abs$_{365}$ (Mm$^{-1}$) | 5.6±4.9 | 10±7.4 | 4.5±4.5 | 1.2±0.21 | 7.2±4.1 |
| MAE$_{365}$ (m$^2$ g$^{-1}$ C) | 0.83±0.25 | 0.96±0.19 | 0.78±0.23 | 0.45±0.06 | 0.95±0.21 |
| FI | 1.6±0.10 | 1.6±0.09 | 1.6±0.08 | 1.7±0.07 | 1.7±0.07 |
| BIX | 0.82±0.13 | 0.83±0.14 | 0.74±0.13 | 0.92±0.05 | 0.89±0.07 |
| HIX | 3.4±0.99 | 3.3±1.1 | 3.9±1.1 | 2.5±0.22 | 2.9±0.36 |
| MSOC | | | | | |
| µg C m$^{-3}$ | 6.0±3.4 | 9.2±4.0 | 4.3±2.9 | 3.9±0.86 | 8.1±2.6 |
| AAE (330–400 nm) | 5.2±0.94 | 4.9±0.69 | 5.5±1.1 | 5.1±0.15 | 4.7±0.55 |
| Abs$_{365}$ (Mm$^{-1}$) | 1.7±1.4 | 1.9±1.6 | 1.0±0.99 | 0.72±0.23 | 2.7±1.4 |
| MAE$_{365}$ (m$^2$ g$^{-1}$ C) | 0.26±0.12 | 0.19±0.08 | 0.23±0.11 | 0.19±0.06 | 0.33±0.11 |
| FI | 1.8±0.20 | 1.5±0.20 | 1.8±0.23 | 2.0±0.10 | 1.8±0.06 |
| BIX | 1.2±0.18 | 1.4±0.20 | 1.2±0.19 | 1.3±0.09 | 1.3±0.14 |
| HIX | 2.0±0.59 | 1.3±0.41 | 2.1±0.68 | 1.9±0.17 | 2.1±0.42 |
| [a]Levoglucosan (ng C m$^{-3}$) | 222±485 | 362±438 | 185±654 | 42±16 | 280±185 |

a: described elsewhere (Wang et al., 2020a).

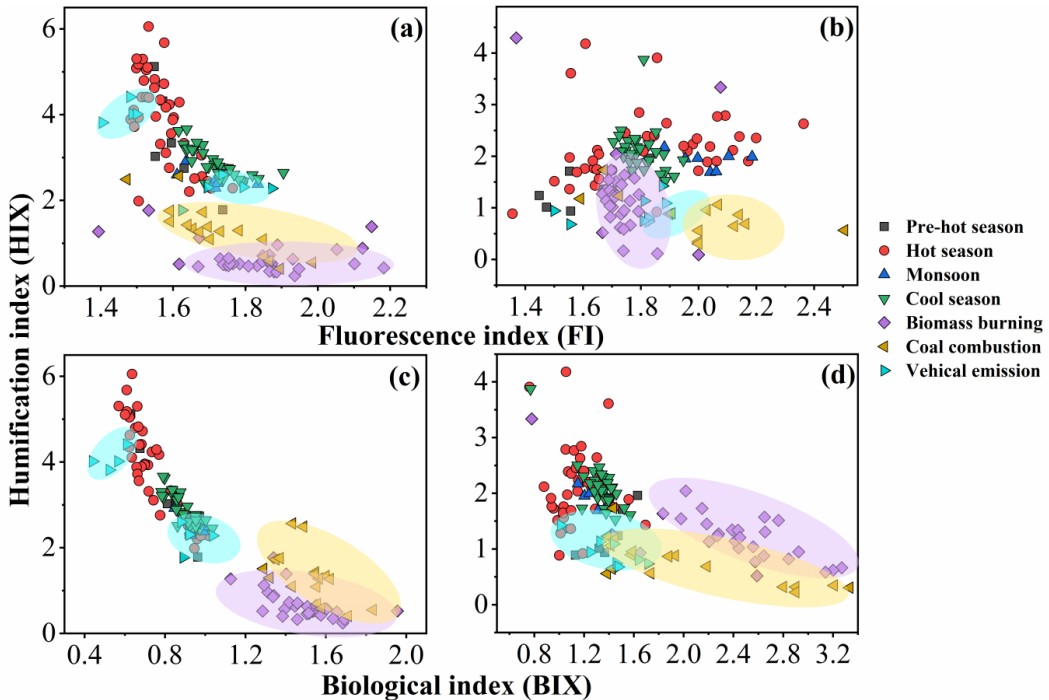


**Figure 2.** Fluorescence index (FI), biological index (BIX), and humification index (HIX) of water-soluble organic

carbon (WSOC, a, c) and methanol-soluble organic carbon (MSOC, b, d) in aerosol samples from Bangkok, Thailand,





as well as source emission samples including biomass burning, coal combustion and vehicle emission which were
encircled by a violet, yellow, and blue region, respectively. Note that the fluorescence characteristic of source samples
was described elsewhere (Tang et al., 2020b), but the fluorescence indices was first reported in this study. Pre-hot
season is from January 18 to February 29, 2016; hot season is from March 2 to May 31, 2016; monsoon is from June
2 to October 30, 2016; cool season is from November 1, 2016 to January 28, 2017.

The BIX and FI were previously proposed as proxies for the contribution of biogenic organic

matter and autochthonous biological activity in natural water, respectively (Fu et al., 2015;Qin et al.,
2018). For example, the FI decreased by up to 20% indicating that the samples appeared increasingly
like "terrestrial" DOM, whereas the BIX increased by up to 37% indicating that the samples became
more "autochthonous" in character (Murphy et al., 2018;Gabor et al., 2014). FI values $\leq 1.4$ correspond
to terrestrially derived organics and higher aromaticity, whereas values $\geq 1.9$ correspond to microbial
sources and a lower aromatic carbon content (Mcknight et al., 2001). An increase in BIX is related to
an increase in the contribution of microbially derived organics, with high values (> 1) shown to
correspond to a predominantly biological or microbial origin of DOM and the presence of organic
matter freshly released into water, whereas values $\leq 0.6$ indicate the presence of little biological
material (Huguet et al., 2009).

The FI and BIX values of the Bangkok aerosol samples are summarized in Table 1 and Fig. 2.

The FI values of the WSOC and MSOC were 1.6±0.10 and 1.8±0.20, respectively, suggesting that
these chromophores are representative of both terrestrially and microbially derived organic matter. The
BIX values of the WSOC and MSOC were 0.82±0.13 and 1.2±0.18, respectively. Almost all BIX
values were greater than 0.6 in the two fractions, suggesting biological or microbial contribution. Lee
et al. (2013) reported that the BIX values of SOA samples averaged 0.6 and increased upon aging. In
addition, the results of our source samples showed that primary biomass-burning and coal-combustion
samples had high FI and BIX values (Fig. 2). These results indicate that these chromophores in
Bangkok were likely freshly introduced or derived from biomass burning and coal combustion. Further,
an increase in BIX in the MSOC in comparison with the WSOC was observed in primary biomass-
burning and coal-combustion samples, consistent with the Bangkok samples. The BIX values were
similar to those in the WSOC in Arctic aerosols (0.6–0.96, mean: 0.72), which were within the extreme
values for the predominance of humic- or protein-like fluorophores (Fu et al., 2015). BIX values
exhibited the opposite trend from HIX values, with low BIX values in the hot season. This may be
explained by a previous study showing that a high BIX appears to indicate little humification (Birdwell
and Engel, 2010). It should be noted that the fluorescence indices (FI, BIX, and HIX) were first applied
for aquatic and soil organic compounds and further extended to the atmosphere due to the similarities
in the properties of organic matter (Graber and Rudich, 2006). However, the values observed for
primary biomass burning and coal combustion in this study differ from with the previously established
fluorescence standards for aquatic environments and soil. Therefore, caution is required when using
these indices to appoint source of atmospheric chromophores (Wu et al., 2021).
**3.3. Optical properties of dissolved BrC.**
Figure 3 shows the variations in soluble OC concentrations and the corresponding light absorption
coefficient at 365 nm (Abs$_{365}$). In general, the Abs$_{365}$ closely tracked the variations in the mass
concentrations of WSOC and MSOC ($p < 0.000$, $R^2 = 0.95$ and $p < 0.000$, $R^2 = 0.75$, respectively) (Fig.
S11), indicating that the portions of BrC in both fractions were considerably stable.

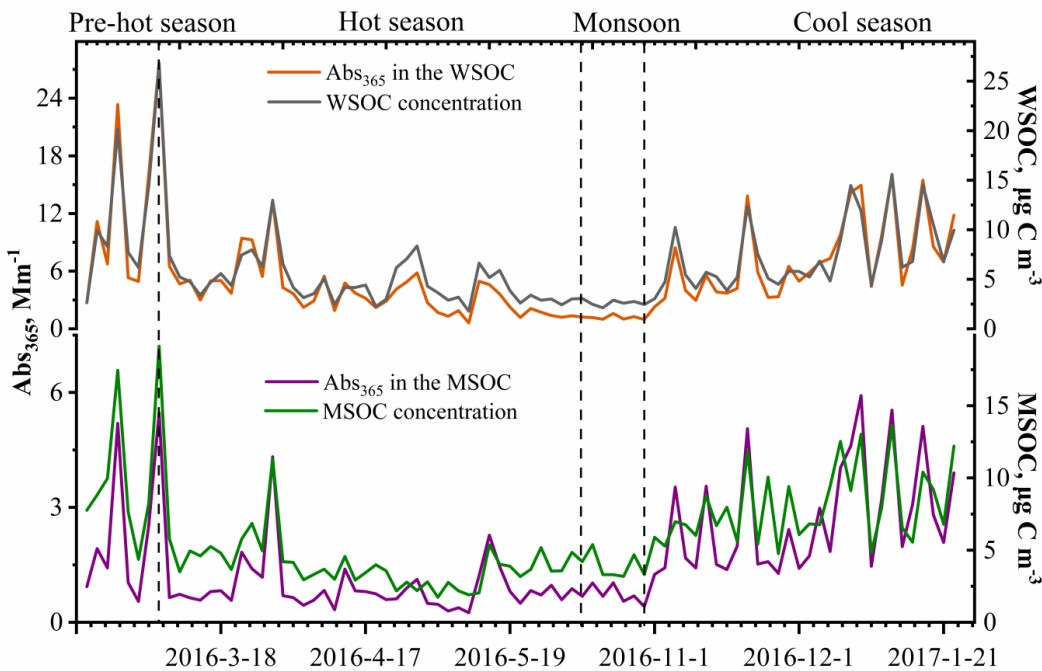


**Figure 3.** Time series plots of water-soluble organic carbon (WSOC) and methanol-soluble organic carbon (MSOC)
concentration (µg C m$^{-3}$) and water- and methanol-extract light absorption coefficient at 365 nm (Abs$_{365}$) (Mm$^{-1}$) in
the aerosol samples from Bangkok, Thailand during 2016–2017.
The absorption Ångström exponent (AAE) and mass absorption efficiency (MAE) are important
optical parameters reflecting the spectral dependence and light absorption ability of BrC, respectively.
The magnitude of the AAE reflects the differences in BrC source and atmospheric processes (Lack et
al., 2013). Typically, the AAE value is close to 1 when light absorption is dominated by soot
(Kirchstetter et al., 2004), roughly 1–3 for simulated biomass-burning aerosols (Hopkins et al., 2007),
and up to 6–7 for water-soluble HULIS in biomass burning-impacted aerosols (Hoffer et al., 2006).



The AAE values of the WSOC and MSOC between 330 and 400 nm in this study were up to 5.1±0.68
and 5.2±0.94 (Fig. 4), respectively, indicating strong wavelength dependence in the light absorption
capability. These high values show that BrC tends to absorb more solar irradiation over ultraviolet
wavelengths, which is comparable to BC absorption as shown in Fig. S12. These observations indicate
that BrC has important impacts on photochemical reactions in the atmosphere (Barnard et al., 2008).
The AAE values in this study are similar to those of water-soluble BrC over biomass burning-impacted
regions, such as Beijing (Mo et al., 2018;Yan et al., 2015) and Guangzhou (Liu et al., 2018), but lower
than those of aerosols from simulated biomass-burning and coal-combustion experiments (Fan et al.,
2018;Tang et al., 2020a;Li et al., 2018). However, it should be noted that the BrC AAE varies in the
atmosphere. Dasari et al. (2019) reported that AAE values of water-soluble BrC increase continuously
due to photolysis of chromophores and atmospheric oxidation during long-range transport over the
Indo-Gangetic Plain (IGP). In addition, pH changes can cause the absorption spectra of some BrC
species to shift to longer wavelengths upon deprotonation, decreasing AAE values (Mo et al., 2017).

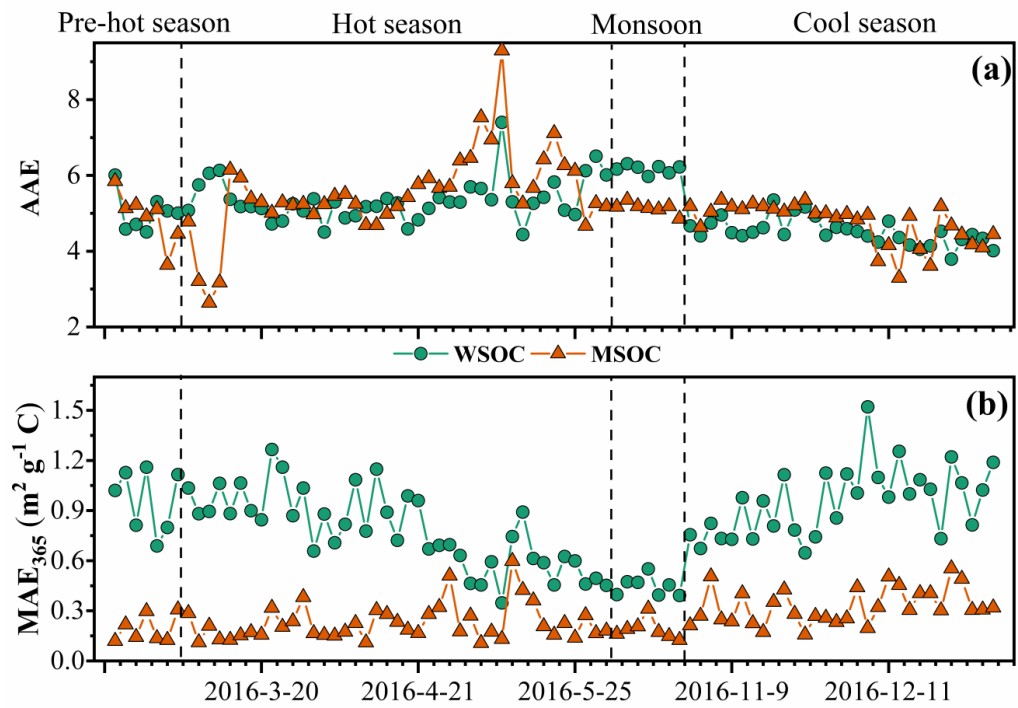


**Figure 4.** Time series plots of Absorption Ångström exponent (AAE, a), the mass absorption efficiency at 365 nm
($MAE_{365}$, b) in the water-soluble organic carbon (WSOC) and methanol-soluble organic carbon (MSOC) in aerosols
samples from Bangkok in Thailand during 2016–2017.



The MAE at 365 nm ($MAE_{365}$) of the WSOC was $0.83\pm0.25$ $m^2$ $g^{-1}$ C, which was higher than that
of the MSOC ($0.26\pm0.12$ $m^2$ $g^{-1}$ C), indicating that more water-soluble BrC with stronger light
absorption capability could be extracted with ultrapure deionized water, whereas water-insoluble BrC
is characterized by lower light absorption capability over Bangkok. These results were consistent with
those from vehicular exhaust samples in our previous study, where $MAE_{365}$ values of the WSOC
($0.71\pm0.30$ $m^2$ $g^{-1}$ C) were higher than those of the MSOC ($0.26\pm0.09$ $m^2$ $g^{-1}$ C) (Tang et al., 2020b).
Opposite results have been shown for primary biomass burning and coal combustion (Tang et al.,
2020b). Wu et al. (2020b) reported that the MAE values of MSOC are higher than those of WSOC in
summer, whereas the situation is reversed in winter. As not all water-insoluble components can be
extracted with methanol, the observed light absorption by MSOC would therefore likely reflect the
lower limit. Table S3 shows a comparison of the MAE values of Bangkok aerosols with those of other
regions, indicating a medium light absorption capacity. The $MAE_{365}$ values of the water-soluble
fraction in this study were comparable to those of Nanjing (Chen et al., 2018), Guangzhou (Liu et al.,
2018), and Beijing in summer (Yan et al., 2015), but lower than those of $PM_{2.5}$ from Singapore (Adam
et al., 2020), $PM_{10}$ from Godavari, Nepal, in the pre-monsoon season (Wu et al., 2019), and smoke
particles from biomass burning and coal combustion (Park and Yu, 2016;Fan et al., 2018;Tang et al.,
2020b). Lower $MAE_{365}$ values of both fractions were observed in the monsoon season than in the non-
monsoon seasons, likely due to the heavy monsoon rains that effectively remove soluble gases and
aerosols (Lawrence and Lelieveld, 2010) and/or reduce biomass-burning activity (levoglucosan level
in Table 1). A previous study reported similar findings in the USA in that the $MAE_{365}$ was
approximately three-fold higher in biomass burning-impacted samples than in non-biomass burning-
impacted samples (Hecobian et al., 2010). Another study in the central Tibetan Plateau highlighted that
BrC emitted by biomass burning has stronger light absorption capability than does secondary BrC
formed in the atmosphere (Wu et al., 2018). On the Indo-China peninsula, Bangkok receives 99% of
the fire-derived aerosols from December to April (Lee et al., 2017), which may explain the high
absorption levels in the non-monsoon seasons.
**3.4. Chromophores responsible for BrC light absorption.**
EEM analysis enables the probing of the chemical structure of DOM because of its ability to
distinguish among different classes of organic matter (Wu et al., 2003). Generally, BrC absorption is
related to the chromophores within it and is susceptible to change with variations in chemical
properties, e.g., oxidation level (Mo et al., 2018), degree of unsaturation (Jiang et al., 2020), molecular
weight (Tang et al., 2020b;Di Lorenzo et al., 2017), functional groups (Chen et al., 2017b), molecular
composition, etc (Song et al., 2019;Lin et al., 2018). The fluorescence intensity of each EEM





component was shown to be associated with light absorption indices, such as $MAE_{365}$ and AAE, of
HULIS in controlled crop straw-combustion experiments (Huo et al., 2018). As a linear relationship
between organic matter concentration and fluorescence intensity can be assumed for very dilute
samples due to the IFE (Murphy et al., 2013), we have corrected our fluorescence data for IFE using
absorbance to enable "clean " correlation analysis (as shown in Fig. S13 a, b). The linear regression
slopes in the scatter plots of $Abs_{365}$ versus WSOC or MSOC could mathematically represent the
average MAE values of WSOC or MSOC at 365 nm, respectively (Fig. S11 a, b). The phenomenon
indicates that both fluorescence and $Abs_{365}$ data point to similar relationships between sources or
chemical processes with organic matter concentrations, and therefore, we attempted to link the
fluorescence results to BrC absorption. It should be noted that light-absorbing substances in
atmospheric particulate matter are not necessarily all fluorescent, such as nitrophenol compounds,
which are a type of BrC commonly found in the atmospheric particulate matter; however, there is no
strong fluorescence signal with which to scan the nitrophenol standards (Chen et al., 2019a).
We used MLR to explore the relationship between the fluorescence intensities of chromophores
and $Abs_{365}$. During MLR, insignificant fluorescent components were excluded from the regression
using a stepwise screening process to avoid overfitting ($F_{inclusion}$: $p < 0.05$; $F_{elimination}$: $p > 0.10$). The
MLR statistical metrics are listed in Tables S4 and S5. For the WSOC fraction, a revised model
(regression 3) equation was used with an adjusted $R^2$ of 0.995. The final optimized equations were
$Abs_{365} = 0.765 \times P4 + 0.051 \times P2 + 0.091 \times P7$, for the WSOC fraction, and $Abs_{365} = 0.238 \times C4$ for
the MSOC fraction (Table S5). The model errors for water-soluble and methanol-soluble $Abs_{365}$ were
−5.5%–64% and −34%–58%, respectively. The predicted $Abs_{365}$ values fit the measured values well
(Fig. 5, slope = 0.99 and 0.95, and $R^2$ = 0.99 and 0.94 for WSOC and MSOC, respectively).

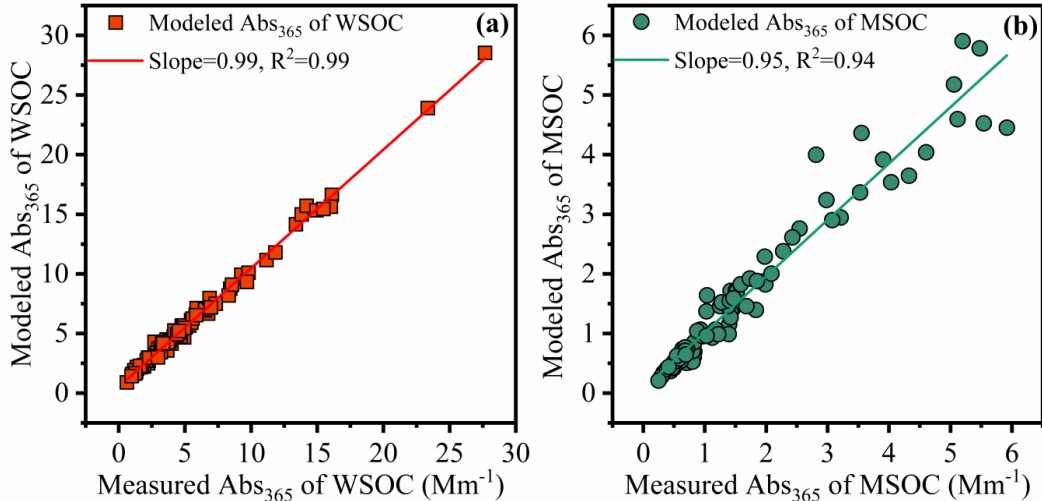


**Figure 5.** Linear correlation analysis between modeling $Abs_{365}$ using multiple linear regression (MLR) analysis and measured $Abs_{365}$ in the water-soluble organic carbon (WSOC, a) and methanol-soluble organic carbon (MSOC, b) in aerosols samples from Bangkok in Thailand during 2016–2017, respectively. Note that the fluorescent intensities of parallel factor (PARAFAC) model results (fluorescent components) were used as variables in MLR analysis.

For water-soluble BrC, the P4 component had the largest coefficient with $Abs_{365}$, which was much higher than those for P2 and P7. The C4 component had the largest coefficient with $Abs_{365}$ for methanol-soluble BrC. These results indicate that the light absorption by BrC is more dependent on chromophores with longer emission wavelengths (P4 and C4). These characteristics also indicate that the strongly absorbing substances in BrC probably originate from large conjugated electron functional groups or include donor and acceptor molecules for charge-transfer interactions (Del Vecchio and Blough, 2004;Cory and McKnight, 2005). Kellerman et al. (2015) reported that these components are highly aromatic and oxygen-rich with high apparent molecular weight. These important findings highlight that larger chromophores may be the most persistent BrC species in the atmosphere and hence exert the greatest influence for perturbing the global radiative balance.

To further interpret the BrC source profiles as real-world TSP sources, we examined 84 (minus one missing value) TSP samples from Bangkok using the US EPA PMF5.0 model. All samples were merged together to form an 84 × 30 dataset (84 samples with 30 species). The initial data of positive matrix factorization input were from our previous study (Wang et al., 2020a). We further added $Abs_{365}$ values of WSOC and MSOC, and the fluorescence intensities (in RU) of P2, P4, P7, and C4 components to the model. A seven-factor solution was achieved that provided the most physically reasonable source profiles (Fig. S14), including ship emission, secondary sulfate, dust, land fossil-fuel combustion, sea salt, biomass burning, and industrial emission, consistent with our previous study



(Wang et al., 2020a). Figure S14 also shows the contributions of the above sources to light absorption
at λ= 365 nm, which represent the fraction of BrC for each factor. Biomass burning was found to be
the main source of BrC over Bangkok; 61% and 67% for water-soluble and methanol-soluble BrC,
respectively. These were comparable to previous observations using a similar approach in Xi'an (55%)
(Wu et al., 2020a). Furthermore, the P4 and C4 components, which were more closely associated with
Abs$_{365}$, could be mostly attributed to biomass burning (64% and 66%, respectively) as shown in Fig.
6. Our previous study showed that biomass burning accounted for a considerably large portion (mean:
26%) of the TSP mass concentration in the same samples (Wang et al., 2020a). This result suggests
that biomass burning makes a significant contribution to not only particulate matter but also BrC light
absorption.

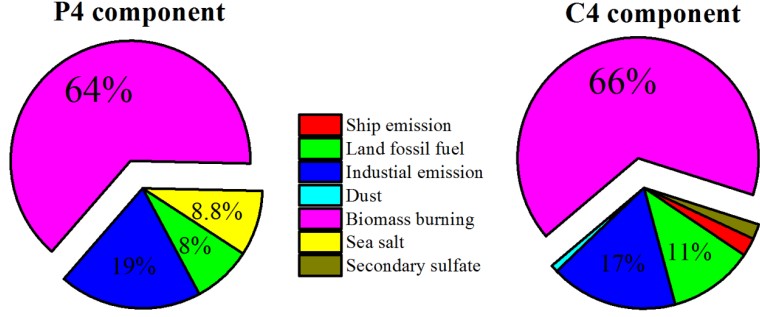

**Figure 6.** Positive matrix factorization derived source apportionment of chromophores P4 of the WSOC and C4 of
the MSOC in TSP samples over Bangkok in Thailand during 2016–2017.
**4.   Conclusions**
This study presents a comprehensive analysis of water- and methanol-soluble chromophores in
aerosol samples over Bangkok in Thailand during 2016–2017. EEM combining with PARAFAC
analysis showed that the identified fluorescent components were humic-like and protein-like
substances but different patterns in the WSOC and MSOC, indicating different chemical compositions.
By adding three-source fluorescence into the original PARAFAC model, we found that chromophores
with longer emission wavelengths in the atmosphere may be due to atmospheric chemical reactions or
"aging" by both bleaching the source chromophores or producing new chromophores. We also suggest
that caution is required when using fluorescence indices to appoint source of atmospheric
chromophores. In addition, more water-soluble BrC with stronger light absorption capability could be
extracted with ultrapure deionized water over Bangkok (0.83±0.25 vs. 0.26±0.12 m$^2$ g$^{-1}$ C), and both
water-soluble and methanol-soluble BrC exhibited a high light-absorption in non-monsoon seasons
due to the influence of biomass burning. The MLR analysis showed that both the light absorption of



BrC at 365 nm in the two fractions was significantly dependent on the special chromophores with
longer emission wavelength that are generally highly aromatic and oxygen-rich with high apparent
molecular weight. Positive matrix factorization model results further showed that biomass burning was
main contributor of these chromophores (up to 60%). In summary, this study provides a new insight
into BrC absorption and sources, which may promote the application of EEM spectroscopy to predict
and model the light absorption of BrC in the atmosphere.
*Data availability.* The data used in this study are available upon request. Please contact Guangcai
Zhong (gczhong@gig.ac.cn).
*Supplement.* The supplement related to this article is available.
*Author contributions.* JT, GZ, JL, and GZ (Guangcai Zhong) designed the experiment. JT and JW
carried out the measurements and analyzed the data. JW and SB organized and performed the
samplings. JT (Jianhui Tang) supported the fluorescence instruments and laboratory. CT and HJ
supported the models. JT wrote the paper. JL, GZ (Guangcai Zhong), YC, YM, BZ, XG, and GZ
reviewed and commented on the paper.
*Competing interests.* The authors declare that they have no conflict of interest.
*Acknowledgements:* This research has been supported by the National Natural Science Foundation of
China (42030715, 41430645 and 41773120), the International Partnership Program of Chinese
Academy of Sciences (grant no. 132744KYSB20170002), Guangdong Foundation for Program of
Science and Technology Research (Grant Nos. 2017BT01Z134, 2017B030314057 and
2019B121205006).

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
