# Peer review of "Measurement report: Long emission-wavelength chromophores dominate the light absorption of brown carbon in Aerosols over Bangkok: impact from biomass burning"

_Atmospheric Chemistry and Physics, 2021_

## Author Response (AR1)

**Title: Measurement report: Long emission-wavelength chromophores dominate the light absorption of brown carbon in Aerosols over Bangkok: impact from biomass burning**
**Manuscript ID: acp-2021-175**

Dear editor,

In compliance with the reviewers' detailed comments, we carefully revised the manuscript. We checked the text and references.

We appreciate the two reviewers for their helpful comments on our manuscript. We considered the detailed comments by the reviewers and responded to their suggestions and questions. For your and the reviewer's easiness to review the manuscript, an annotated manuscript was attached at the end of this file.

We are very sorry for making an error in the PMF model. We mistakenly brought the fluorescence of components of the 145-model into the PMF model, and the correct data should be that of the 85-model. Thus, we have revised it in the revised manuscript (Line 448–458).

We sincerely appreciate your consideration. Look forward to hearing from you soon.

With Best Regards,
Dr. Guangcai Zhong

**Response to Anonymous Referee #1**

**RC- Reviewer's Comments; AC – Authors' Response Comments**

**RC1:** This manuscript by Tang et al. represents a in-depth analysis of chromophores and fluorophores present in filter samples collected for an entire year in Bangkok. The authors use Excitation Emission Matrix, parafactor analysis (PARAFAC) and multiple linear regression (MLR) to provide insights into the contribution of potential sources to light absorbing organic compounds (BrC) in the samples collected. The chemical and data analyses were conducted with cautions. A year-round data from Bangkok serves as a precious case study for the community to understand light-absorbing organic compounds and their climate impact. I recommend publication in ACP after addressing the following minor comments.

**AC1:** Thanks for your recognition of our work and for providing valuable suggestions. We have made revisions following the comments (corrections are marked in the revised manuscript), and the responses are shown below.

**RC2:** Figure 1 - the color scale is not explained. Is it normalized to 1 for the highest intensity among all the factors?

**AC2:** We normalized to 0.1 for the highest intensity among all factors. We have added it as following: The color represents that the intensity was normalized to set the maximum as 0.1. Please see line 205-206 in the revised manuscript.

**RC3:** Figure 3 and related discussion. Although I agree with the authors in that the ratio of Abs365 and WSOC/MSOC is consistent, I also see that Abs365 is enhanced relative to WSOC/MSOC during the non-monsoon seasons. I wonder if the authors can investigate the ratios and discuss whether WSOC is more absorbing during BB-affected seasons?

**AC3:** We did see that the $Abs_{365}$ is enhanced relative to WSOC/MSOC during the non-monsoon seasons. According to Table 1, the concentrations of levoglucosan that are generally regarded as biomass burning tracers were higher in the non-monsoon seasons than in monsoon season, and the ratios of levoglucosan/TSP also exhibited a similar trend (we have added the ratios in the revised manuscript, Table 1). Thus, we infer that the non-monsoon season was more affected by biomass burning. Correspondingly, both the WSOC and MSOC are more absorbing during the biomass burning-affected seasons (Table 1). Please see line 322-326 in the revised manuscript.

**RC4:** Line 387~ I think a little more discussions regarding the MLR results can be helpful for the community. Can the authors conclude that Abs365 in both WSOC and MSOC is dominated by a single factor (P4 for WSOC, C4 for MSOC). Is this result consistent with previous EEM and PARAFAC studies?

**AC4**: We have re-detailed the content of this part and please see line 411-419 in the revised manuscript. In this study, we attempted to build an MLR model to explore the relationship between the BrC absorption and fluorescent chromophores, and we thought that the coefficient (not a constant) in the equation represented the strength of the relationships. Thus, we can conclude that $Abs_{365}$ is dominated by the single factor (P4 for WSOC, C4 for MSOC).

A similar study was conducted. Chen et al. (2019) thought that organic substances may represent the important causes of DTT (dithiothreitol) consumption and may be mainly contributed by light-absorbing materials. They found only two fluorescent components contribute to the DTT activity, almost all of which is attributed to the C7 chromophores with an emission wavelength of 462 nm (99%).

**Reference:**

Chen, Q., Wang, M., Wang, Y., Zhang, L., Li, Y., and Han, Y.: Oxidative Potential of Water-Soluble Matter Associated with Chromophoric Substances in PM2.5 over Xi'an, China, Environ. Sci. Technol., 53, 8574-8584, https://doi.org/10.1021/acs.est.9b01976, 2019.

**RC5:** Related to my previous comment, for WSOC, both P3 and P4 have an excitation maximum at around 365 nm. However, only P4 has a significant coefficient after MLR analysis. Meanwhile, P3 had a negative coefficient. Why is this?

**AC5:** In our MLR analysis, we set the constraints and manually selected regression model 3 for our optimal model. Regression model 4 is a calculation process and was also abandoned. However, the negative coefficient of P3 with $Abs_{365}$ in regression model 4 could be due to the following reasons. First, from the mathematical meaning, in MLR analysis, simple correlation coefficients may not truly reflect the correlation between variables X and Y, because the relationship between variables is complex and they may be affected by more than one variable. Thus, the partial correlation coefficient is a better choice. In this study, to obtain the real correlation, we control the variables P4, P2, and P7 and make a correlation analysis between $Abs_{365}$ and P3. The partial correlation analysis shows that $Abs_{365}$ and P3 have a negative correlation ($r=-0.227$, Table A1), which is consistent with the negative coefficient in regression model 4. However, P3 does have an excitation at around 365 nm. According to the study of Phillips and Smith (2014), they explained that light absorption by organic aerosols is governed by a combination of independent as well as interacting chromophores. They introduce the charge transfer (CT) complexes as a significant source of BrC absorption, formed through the interaction between carbonyl and alcohol moieties in organic molecules, are energetically coupled to one another and form a near-continuum of states that absorb light from 250 to 600 nm, which may contribute up to 50% of light absorption (300–600nm) of the water-soluble fraction. In their experiment, they used $NaBH_4$ to individually reduce carbonyl functional groups in ketones and aldehydes, likely electron acceptors in CT complexes, to the corresponding alcohols. They observed that absorption spectra decrease and fluorescence increase of aqueous solutions after reduction by $NaBH_4$ (Phillips and Smith. 2015). This correlation between fluorescence gain and absorption loss demonstrates that the increased fluorescence is unlikely to result from newly created absorbing species and thus likely results from previously existing species that are more strongly fluorescing after reduction. Therefore, we infer that P3 likely a donor to interact with the other components to form CT complexes, resulting in increased absorption, but decreasing fluorescence due to the loss of functional groups. We hope the above points explain the reason that P3 has a negative coefficient with $Abs_{365}$.

**Table A1.** The partial correlation analysis between $Abs_{365}$ and P3 under the control variables P2, P4, and P7.

| Control variables | | | P3 | $Abs_{365}$ |
|---|---|---|---|---|
| P2 & P4 & P7 | P3 | *r* | 1.000 | -0.227 |
| | | *p* | | 0.040 |
| | $Abs_{365}$ | *r* | -0.227 | 1.000 |
| | | *p* | 0.040 | |

**Referaces:**

Phillips, S. M., and Smith, G. D.: Light Absorption by Charge Transfer Complexes in Brown Carbon Aerosols, Environ. Sci. Technol., 1, 382-386, https://doi.org/10.1021/ez500263j, 2014.

Phillips, S. M., and Smith, G. D.: Further evidence for charge transfer complexes in brown carbon aerosols from excitation-emission matrix fluorescence spectroscopy, J. Phys. Chem. A, 119, 4545-4551, https://doi.org/10.1021/jp510709e, 2015.

**Response to Anonymous Referee #2**

**RC1:** In this manuscript, the authors present a comprehensive study of water- and methanol-soluble chromophores and fluorophores in brown carbon (BrC) from aerosol samples collected year-round in Bangkok, using absorption and excitation-emission matrix (EEM) spectroscopies and numerical methods, including both parallel factor analysis (PMA) and positive matrix factorization (PMF). The selection and preparation of samples and the spectral and factor analyses are all well designed and carefully executed. The observations of chromophores and fluorophores together provide insights into the origin and fate of BrC in the atmosphere. For example, the PMA analysis of EEM observations indicates that atmospheric aging shifts the wavelengths of emission from fluorophores, as primary species react and secondary species form. The PMF analysis indicates that components most associated with absorption at 365 nm are largely emitted from biomass burning. These components are also characterized by long emission wavelengths, suggesting that the constituent molecules incorporate extended conjugated systems or charge-transfer interactions. With these and other impactful implications, the manuscript is suitable for publication in ACP. I have only technical and minor comments for the authors to consider.

**AC1**: We greatly appreciate the reviewer for recognizing the merits of this work and for providing valuable suggestions. These suggestions would help us improve the manuscript.

**RC2:** Line 31 - Please reword "these inferences exhibited a refutation".

**AC2**: Thanks for your suggestions, we have revised it as follows: which exhibited a different characteristic from primary biomass burning and coal combustion results. Please see line 31-32 in the revised manuscript.

**RC3:** Line 80 - Here and throughout the use of chromophore versus fluorophore is sometimes ambiguous. I would argue the claim in this sentence is not true, since many previous studies have explored the relationship between chromophores (i.e., the species that give a material its color) and BrC. If chromophores were to be replaced with fluorophores, or "fluorescent chromophores" as in line 94, the claim is not so problematic.

**AC3:** Thanks for your suggestions. We did confuse "fluorophore" and "chromophore". In the previous studies, for example, Lin et al. (2016, 2017, 2018) have investigated the relationship between chromophores and BrC using the combination of high-performance liquid chromatography (HPLC), photodiode array (PDA) spectrophotometry, and high-resolution mass spectrometry (HRMS). They identified BrC chromophores like polycyclic aromatic hydrocarbons (PAHs), heterocyclic O-PAHs and N-PAHs, nitro-phenols, et. The compounds can regard as chromophores. However, not all chromophores were fluorophores. In the study of Chen et al. (2019), they separately measured the light absorption and fluorescence properties of several nitrophenol standards, and the results showed strong light absorption properties, but no strong fluorescence signal was observed. Thus, we have replaced chromophores with "fluorescent chromophores" throughout the revised manuscript.


**RC4:** Line 112 - Include units of electrical resistivity.

AC4: We have added the units of electrical resistivity (resistivity of > 18.2 MΩ) in the revised manuscript. Please see line 114 in the revised manuscript.

**RC5:** Line 113 - Replace "GFFs" with "QFFs".

**AC5**: Thanks for your revision, we have revised it.

**RC6:** Line 213 - Please consider rephrasing the sentence beginning with "Although one exceptional component was detected..." The meaning is not clear to me.

**AC6**: In this part, we wanted to express that one additional fluorescent component was identified in the new 145-model (the model contains the total EEMs of 60-sources samples and 85-Thailand TSP samples) for WSOC and MSOC fraction comparing with the 85-model (the model only contains the EEMs of 85-Thailand TSP samples), respectively. Now, we have revised the sentence as follows: It should be noted that one additional fluorescent component was identified each for the WSOC and MSOC fractions in the new 145-model, respectively, but these components were only highly characterized by source emission samples. Please see line 219-224 in the revised manuscript.

**RC7:** Line 335 - Mention the range of measured pH values.

**AC7**: The pH values of WSOC fraction for all the samples were within the range of 5–7, generally thinking it didn't affect the absorbance according to a prior study (Chen et al., 2016). We have added it, please see line 350-351 in the revised manuscript.

[revised manuscript text omitted]

---

## Author Response (AR2)

**Response to editor**

Journal: ACP

Title: "Measurement report: Long emission-wavelength chromophores dominate the light absorption of brown carbon in Aerosols over Bangkok: impact from biomass burning"

Author(s): Jiao Tang et al.

MS NO.: acp-2021-175

Dear editor,
    We are pleased for our work to be accepted on ACP. Thanks.

With Best regards,
Dr. Guangcai Zhong